# Development and Evaluation of Nursing Clinical Practice Education Using M-Learning

**DOI:** 10.3390/healthcare12020206

**Published:** 2024-01-15

**Authors:** Sungeun Kim, Mihae Im

**Affiliations:** 1Department of Nursing, Daedong College, Busan 46270, Republic of Korea; tokaki3@naver.com; 2Department of Nursing, Daegu Hanny University, Gyeongsan-si 38610, Republic of Korea

**Keywords:** learning, satisfaction, clinical competency, problem solving, nursing students

## Abstract

This study aims to develop and evaluate the effectiveness of nursing practice education using mobile learning or m-learning for nursing students. A nonequivalent control group post-test design was used. Overall, 42 nursing students participated in the study. A three-week nursing practice education program was developed using the Analysis, Design, Development, Implementation, and Evaluation (ADDIE) model. The course was implemented on the basis of Gagne’s nine instructional situations. The findings demonstrated improvements in clinical competency (t = 7.44, *p* < 0.001) and problem solving (t = 2.29, *p* = 0.028). Accordingly, the study recommends introducing m-learning in nursing practice education using tablet PCs, as part of a newer nursing practicum training strategy that takes into account the factors identified in this study. It is also suggested that a continuous m-learning approach and development plan for nursing students be prepared to achieve technically advanced nursing practice education.

## 1. Introduction

The objective of nursing education is to enable students to acquire practical skills through theoretical knowledge and experience to choose a desirable nursing process in diverse clinical situations [1]. Clinical practice is particularly beneficial for nursing students to acquire overall nursing knowledge through training processes and methods that can be integrated, applied, and utilized in clinical situations [2]. However, considering that the subjects that nursing students encounter during clinical practice are human beings, mistakes are not acceptable [3]. Additionally, opportunities for nursing students to perform direct nursing in clinical practice are declining because subjects also prefer career nurses [4]. In addition, the difficulties that students experience in new environments, including handling the latest equipment and their relationships with patients and other healthcare providers, make them lose confidence in their nursing practice [3,4]. Nursing students mainly engage in clinical practice focused on observation; therefore, when they enter the actual clinical field, they experience difficulties owing to the difference between the knowledge they acquired at school and nursing work in actual clinical settings [3,4,5].

Department heads in the clinical field reported that new nurses lacked basic knowledge and adaptability in the nursing field [6], and that they had theoretical knowledge but did not apply it appropriately in practice [5,6]. Therefore, a significant number of new nurses presented the problem of being unable to take responsibility for nursing care for patients immediately after employment due to a lack of preparation for practical work [6,7]. Therefore, there is a need for nursing educators to prepare teaching strategies that consider the key elements of teaching and learning so that nursing students can perform nursing activities in clinical settings based on sufficient understanding.

In nursing education, simulators, standardized patients, and action learning are introduced to improve students’ practical skills, allowing them to have clinical field experience rather than education conducted through traditional classroom lectures [8]. Students nevertheless have trouble expanding their theoretical knowledge and improving practical skills during practice because of time and space constraints and gaps in theoretical learning and practice [9,10]. Therefore, in clinical practice, education that connects nursing knowledge with practice must be systematically conducted [6], and specific teaching and learning content, methods, and media must be developed to supplement this [11].

With the rapid growth of the Internet of Things in the wake of advancements in information and communication technology, a ubiquitous environment connecting the real world and cyberspace has been created, including learning fields using u-learning, e-learning, and m-learning [12]. Additionally, recent learners are in the “one-person, one-mobile” era and are using tablet PCs and smartphones in various ways for learning and everyday activities [13]. Educators must also accept current trends and changes in learners and incorporate them into the new educational environment [12,13]. In particular, in the case of clinical practice education, since nursing students leave school and learn through practical training in hospitals, class management and interaction with instructors are increasingly required compared to a general learning environment [6,14]. M-learning can potentially address diverse inquiries and demands.

The learner becomes the subject of m-learning, which helps expand knowledge and stimulate interest. It also has a positive impact on self-directed learning [15]. Constructivism emphasizes learning that focuses on the learner [16]. Constructivist learning theory focuses on direct performance based on a learner’s understanding [17]. When applying m-learning to clinical practice education, constructivist learning theory is deemed a desirable framework [16,17].

However, in m-learning, considering that an individual learns in a virtual space, cognitive presence or learning immersion may be lowered and the expected learning outcome may not be achieved [18]. To ensure that learners achieve the learning effects expected of educators, appropriate instructional design must be supported, along with the composition of educational content that reflects clinical practice.

Nursing clinical practice education may not be capable of providing effective education to nursing students because of discrepancies between theory and knowledge, lack of advanced clinical experience among clinical practice educators, or problems caused by the delegation of student guidance [1,10]. In addition, considering the current status of practice for patients with different clinical practice sites and severity [2,5], research on the role and teaching efficiency of clinical practice educators is needed, and a systematic and qualitative instructional design should be prepared for this [6].

Therefore, this study aims to develop an m-learning program using tablet PCs and apply an instructional design as a teaching strategy to effectively educate nursing students to improve their clinical practice skills. In addition, by testing the effectiveness of m-learning practice education, this study intends to use basic data for nursing practice education strategies. The specific purposes are as follows: (i) develop m-learning nursing practice education content using tablet PC, and (ii) test the effectiveness of the program on nursing students.

## 2. Materials and Methods

### 2.1. Study Design

This study was designed with a nonequivalent control group post-test design (Table 1).

### 2.2. Participants

The participants were third-year nursing students attending D University in South Korea. Using purposive sampling, students were sampled from a university where two classes alternately teach theory and practice for three weeks. Since there is a possibility of spread among the research subjects of each group during the period of theoretical lectures on campus, the subjects of this study were selected as students who were assigned to practice in the same subject during the same period. The specific selection criteria of this study were as follows: first, 3rd-grade nursing students enrolled in D University; second, students who were taking adult nursing clinical practice; and third, students who agreed to participate in this study. There were concerns about the disadvantages they would receive from the professor, so each student was given a randomly assigned number, and the teaching assistant was selected as the recipient by drawing lots. Subsequently, the purpose and process of the study were explained, and students who agreed to participate were chosen as subjects. Moreover, they were informed that participating in this study would not affect credit for major courses.

The required sample size was calculated using G-3.1.2. A total of 42 participants were required for this study, with an effect size of 0.80, which was determined based on a previous study [19], a significance level of 0.05, and a statistical power of 0.80. Considering a 10% attrition rate, 46 students were recruited for the treatment and control groups. In the control group, two students dropped out, citing personal issues (8.7% dropout rate). In the experimental group, one student each dropped out for personal issues and to take the semester off school (8.7% dropout rate). Therefore, 42 participants from each of the experimental and control groups were included in the final analysis.

### 2.3. Ethical Considerations

This study was approved by the Institutional Review Board of I University (2-11041024-AB-N-01–20160222-HR-364). All participants were provided with a form explaining the background and purpose of the study, survey content, benefits of participation, confidentiality, and right to withdraw from the study. Traditional nursing care was provided to the control group during the study period, and m-learning nursing practice education was provided in the next semester. Gift certificates were provided to participants as a gesture of gratitude.

### 2.4. Measurements

#### 2.4.1. Learning Satisfaction

Learning satisfaction was measured using a scale developed by Lee [20], and modified by Seong [21]. This tool gauges satisfaction levels concerning the appropriateness of learning content, learning outcomes, and the adequacy of learning evaluation. The scale consisted of 20 questions rated on a five-point Likert scale (1 = strongly disagree to 5 = strongly agree). Higher mean scores indicated greater learning satisfaction. Cronbach’s α was 0.80 in the previous study [21] and 0.95 in this study.

#### 2.4.2. Clinical Competency

Clinical competency was assessed using a scale initially developed by Lee [22], and subsequently modified by Gweon [23]. The scale comprised 45 questions organized into five factors: 12 questions for “nursing process”, 13 questions for ”nursing skill”, 8 questions for “cooperation”, 3 questions for “communication”, and 9 questions for “professionalism”. Respondents used a four-point Likert scale ranging from 1 (strongly disagree) to 4 (strongly agree). Elevated mean scores indicated greater clinical competency. Cronbach’s alpha for the scale was 0.95 during its development [22] and increased to 0.96 in this study.

#### 2.4.3. Problem-Solving Ability

Problem-solving ability was measured using a scale originally developed by Lee [24], and subsequently modified by Woo [25]. This measurement is based on students’ self-reported scores related to their typical problem-solving skills, encompassing 5 questions for “discovering the problem”, 5 questions for “defining the problem”, 5 questions for “devising a solution to the problem”, and 5 questions for “executing the solution” with a five-point Likert scale (1 = strongly disagree to 5 = strongly agree). A higher mean score indicated higher problem-solving abilities. Cronbach’s α was 0.89 in a previous study [25], and it reached 0.92 in this study.

#### 2.4.4. Self-Directed Learning

This was measured using the “Self-Directed Learning Scale” developed by Lee [26] for college students and adults. The tool comprises three competency elements and eight sub-elements: learning plan, learning execution, and learning evaluation. Sub-elements for each competency element include learning plan-learning needs diagnosis, goal setting, resource identification for learning, learning execution-basic self-management ability, learning strategy selection, continuity of learning execution, learning evaluation-effort attribution, and self-reflection. The scale consisted of 45 questions, each rated on a five-point Likert scale (1 = strongly disagree to 5 = strongly agree). Higher mean scores indicated greater self-directed learning. Cronbach’s α was 0.94 during its development [26], and remained high at 0.93 in this study.

### 2.5. Development of Nursing Clinical Practice Education with M-Learning

The program was developed based on the analysis, design, development, implementation, and evaluation (ADDIE) model [27]. In the implementation stage, class was performed according to Gagné’s nine events of instruction [28]. The control group was evaluated using traditional educational methods. In the evaluation stage, according to Kirkpatrick’s evaluation model, the first stage, learner satisfaction, and the second stage, clinical competency, problem-solving ability, and self-directed learning, were measured (Figure 1).

#### 2.5.1. Analysis

Focus group interviews were conducted to understand the perceived clinical practice education needs of five nursing students at D University in order to improve their clinical practice skills [29]. The subjects participating in the focus group interview were fourth-year students who had completed the entire clinical practice curriculum in the nursing department and had undergone the same curriculum as the research subjects attending this study. Additionally, the educational requirements of clinical educators and nursing practitioners were analyzed for seven clinical educators and two nurses with more than five years of clinical experience. Through interviews, we created a plan for the content of the m-learning program and confirmed what was expected from clinical practice education.

A literature review was performed on the clinical practice learning contents, educational needs, instructional design, teaching and learning strategies, and media operation. Searches were conducted using five databases in Korea, including Medical Research Information Center (MedRIC), KoreaMed Synapse, Korean Studies Information Service Systerm (KISS), National Digital Science Library (NDSL), and Research Information Sharing Service (RISS). A total of sixteen studies were analyzed, confirming the necessity of a systematic teaching and learning method for efficient clinical practice education. Additionally, it was confirmed that an effectiveness evaluation was required through an objective self-report questionnaire administered to learners after education.

The level of prior learning, practice paper, and practice time of the students were identified for learner analysis. Analysis of the learning environment included identifying spaces, media, and equipment facilities for learning. Consequently, approximately 60 tablet PCs were already set up for m-learning, and a curriculum was created that considers student accessibility and efficiency. A Tablet PC was used to organize and store educational programs by week, and the programs could be downloaded and stored at any time.

Finally, the task and job analyses were conducted. In addition to reviewing the theoretical content of prior learning, the contents of the analysis of learners and their learning needs were referred to. Additionally, the current learning objectives, outcomes, and connections were reviewed to incorporate them into the program’s structure.

#### 2.5.2. Design

Design-specified learning goals, program structures, and series were completed by synthesizing the results of the analysis process. Furthermore, plans were made to implement teaching strategies and media use. The specific tablet PC usage plan was composed before and after the practice, as shown in Table 2.

To confirm the effectiveness of the program, we conducted an evaluation using Kirkpatrick’s evaluation model [30]. In addition, detailed performance goals according to the learning goals and learning outcomes were specified. Based on this, the learning content and teaching guidance schedule were created according to a three-week practice schedule.

A post-evaluation scenario for clinical performance assessment by a research assistant was created, and a detailed plan and agreement on the evaluation method were reached with the research assistant. Subsequently, a preliminary simulation was conducted to ensure the reliability of the evaluation.

Additionally, the program was explained in advance to clinical field educators so that students could learn using tablet PCs for more than 90 min every day. Moreover, weekly learning content was announced, and learners were able to adjust their assignment schedules. An overall plan for the program operations was also developed to ensure that the curriculum was adequately provided to the experimental group, as planned by the researcher.

#### 2.5.3. Development

The content of clinical practice education using the m-learning program consisted of providing examples through cases for each topic, as well as additional explanations and analyses. The students were given specific tasks and problems to apply after learning, and they went through the stages of “example–explanation–application”. The weekly learning content is presented in Table 3.

The teaching and learning guidance plan linked the generally operated processes of “introduction”, “development”, and “organization” to learners’ internal cognitive processes of “preparation for learning”, ”acquisition and performance of information and technology”, and “regeneration and transfer of learning” during Gagné’s nine instructional events (Table 4). The teaching and learning guide plan was reviewed, revised, and supplemented by nursing education experts. Pilot education was conducted for three fourth-year nursing students; their opinions were collected, and a revision process was conducted.

A scenario for evaluating clinical competency was developed with one nursing professor and two nurses with master’s degree, having more than five years of clinical experience. There are a total of three cases, and cases were developed in connection with the contents of all clinical competency evaluation items.

Clinical competency was evaluated by a research assistant to maintain the internal validity of the study. A preliminary evaluation was also conducted using the developed scenario and evaluation criteria to confirm the validity of the evaluation, and then final revisions and supplements were made.

#### 2.5.4. Implementation

The clinical practice of the research subjects lasted for three weeks, and each of the experimental and control groups was categorized into separate groups, and the clinical practice period and class in the college period were alternated. Therefore, the experimental and control groups were able to study alternately on campus and at the clinical practice site, respectively, to prevent contamination among subjects.

For three weeks, from 11 April to 29 April 2016, the control group was involved in traditional adult nursing clinical practice. Their clinical practice education included 135 h at the hospital, one hour of orientation training, three hours of field guidance training, and 12 h of conference training. After the practice ended, there was an hour of reflection on campus.

The experimental group was provided nursing practice education using m-learning for three weeks, from 2 May to 20 May 2016. The practical instruction time was the same as that of the control group, and each participant was provided a tablet PC. During the orientation before practice, we taught them how to use the tablet PC and provided guidance on how to apply and utilize the program through regular learning. The educational content developed was utilized during conferences, and log writing and blank-filling were used to assess student learning.

The clinical practice education using m-learning followed Gagné’s nine instructional events (Table 4). The first step was “gaining attention” during the learning preparation phase. At the beginning of each class, the researcher delivered an announcement, presented a clinical situation using a video, or conducted an activity to evaluate experiences gained during practice. The second step involved “setting the class expectations”. The instructor communicated the post-class learning outcomes and detailed learning objectives connected to the m-learning program’s educational content. Understanding and motivation were fostered by illustrating nursing tasks observed and performed during clinical practice. The third step, “Stimulation recall of prior learning”, encompassed reviewing individually learned m-learning, checking assignment performance results, providing feedback, and verifying learning through practice logs. In the fourth stage, “selective perception”, instructors guided learners on what to learn, provided instructions on related m-learning program education and clinical practice, and encouraged students to explore similar learning cases independently.

The fifth step, “encoding of meaning”, involved reinforcing theoretical learning contents by linking them to nursing activities experienced in clinical practice. The m-learning program facilitated methodical extended learning on data collection and analysis of clinical cases, nursing activity skills, prioritization, and treatment. The sixth stage, “playback and response”, enabled learners to practice independently or recall memories through assignments, practice problems, relearning videos, and using models. The seventh step, “reinforcement”, included providing feedback on the success and accuracy of learning and prompting self-evaluation. Instructor feedback, problem-solving guidance, discussion, peer evaluation, and on-site guidance were utilized. The eighth step, “evaluation”, assessed the level of understanding through the presentation of similar cases or problems for self-evaluation, followed by a debriefing process in a conference meeting. Lastly, the ninth step, “generalization”, confirmed the connection with theoretical learning content based on clinical practice experiences completed using a quiz and an integrated summary.

#### 2.5.5. Evaluation

The evaluation of this training was based on Kirkpatrick’s evaluation steps. General characteristics and learning satisfaction were surveyed during the first stage of evaluation. The initial stage of the evaluation involved self-reporting at the end date of adult nursing practice in the experimental and control groups. To assess whether learners had learned what was intended, Kirkpatrick’s second stage of evaluation involved measuring what they had learned. The experimental and control groups underwent clinical performance evaluations based on three scenarios under the supervision of one research assistant on the last day of practice and self-reported evaluations of problem-solving ability and self-directed learning.

### 2.6. Data Analysis

Data analysis was conducted using SPSS version 22.0. Homogeneity testing for general characteristics was performed using the independent *t*-test, chi-square test, and Fisher’s exact test. After checking the normality of the variables, a comparison of the changing variables between groups was performed using the *t*-test.

## 3. Results

### 3.1. Homogeneity Test for General Characteristics

There were no significant differences in the general characteristics between the two groups (*p* > 0.050) (Table 5).

### 3.2. Effects of Nursing Practice Education Using M-Learning

There were no significant differences in learning satisfaction between the groups (t = 0.37, *p* = 0.711). In terms of clinical competency, the mean score of the experimental group was significantly higher (2.87 ± 0.33) than the control group (2.12 ± 0.32) (t = 7.44, *p* < 0.001). The mean scores for all factors of clinical competency were also significantly higher in the experimental group. For problem solving, the mean score of the experimental group was significantly higher (3.74 ± 0.41) than the control group (3.44 ± 0.43) (t = 2.28, *p* = 0.028). Lastly, the mean score of self-directed learning was not significantly higher (3.39 ± 0.44) than the control group (3.18 ± 0.32) (t = 1.68, *p* = 0.101) (Table 6).

## 4. Discussion

The development of a tablet PC-based m-learning program was used as a nursing practice education strategy to improve the clinical practice skills of nursing students, and its effectiveness was confirmed in this study. We will divide this discussion into two parts: the process of development m-learning program and its effectiveness.

### 4.1. Development of Nursing Education Using M-Learning

In this study, the tablet PC was selected as the medium of education, considering that it improves accessibility by combining the portability of a cell phone with the functions and advantages of a laptop [31]. Clinical practice involves students departing from school and classrooms to learn in hospitals. To effectively manage students and provide them with feedback and visual educational materials, m-learning education methods are needed [32]. Therefore, the strength of this study was believed to be the use of a tablet PC that can be accessed anytime, anywhere, and has mobility and visual effects as a tool to compensate for the limitations of the clinical practice education environment.

The use of new media in student education requires a change in the students’ roles, which emphasizes understanding and experience [33]. During conferences and on-site guidance, the instructor used m-learning to link theoretical learning content with practice and presented various examples and usage plans to emphasize practical aspects. In addition, instant feedback was provided using m-learning, enabling the learned content to be reobserved in the field. Concurrently, learning through “empirical knowledge” was stimulated by having learners select and observe cases on their own and analyze nursing activities based on theoretical learning contents. Consequently, it was possible to link theoretical learning and practice pursued in clinical practice education. In addition, the instructor was faithful to the role of a facilitator, and the learner was the subject of learning and achieved self-directed learning, which is believed to reflect constructivist learning theory.

Each step of the experiential learning model required to complete the learning was included in the nursing practice education strategy developed in this study. Various situations have been encountered to directly experience health problems in the clinical field during specific stages. The ADDIE model was employed to develop and apply an m-learning program to help students reflect on systematic analysis and design during the reflective observation stage. In addition, it underwent an abstract conceptualization step that led to generalization and transfer by introducing a teaching design that applied Gagné’s nine instructional events to link nursing knowledge and practical experiences. Finally, in the active experimental phase, Kirkpatirick’s four levels of evaluation were applied to test the effectiveness of the strategy. The elements applied at each stage were judged to be methods that can be applied evenly to most nursing students by ensuring the feasibility of strategy development based on scientific evidence.

The content of the m-learning program reflects the needs of field leaders, clinical nurses, and students with experience in the clinical field. The m-learning program developed in this study was judged to reflect the nursing work in the field of clinical practice as realistically as possible. Furthermore, the design of the program organization, composition, and teaching methods so that the composition of the m-learning program can be organically linked to existing clinical practice education programs, seems to be an approach that considers various contextual aspects when introducing new teaching and learning methods.

This study applied Gagné’s nine instructional events. In a previous study that applied Gagné’s class situation, it was said that a wide range of learning activities could be achieved by understanding the concept of learning content for application [34,35]. Moreover, through this process, learners can utilize the learned content at a higher level, leading to an increase in transfer [34,35,36]. This study also helped organize practical training by describing specific questions, related materials, expected learner responses, and time required based on Gagné’s nine instructional events. Therefore, it was beneficial to develop and apply Gagné’s instructional events in teaching and learning guidance to the nursing practice education attempted in this study.

By applying the Kirkpatrick evaluation model, which emphasizes a methodical evaluation centered on performance and results, the learning effects at each stage were identified and analyzed. This analysis serves as a foundation for enhancing the development of nursing practice education strategies. The Kirkpatrick evaluation model delineates the evaluation of learning effects across four stages: reaction, learning, behavior, and result. The initial stage involves evaluating the learner’s response, primarily to verify satisfaction or personal opinions. Learning satisfaction in this study was used to confirm Kirkpatrick’s first-stage evaluation as a subject response evaluation.

To evaluate whether the learner has achieved the intended learning, the second step assesses their knowledge, skills, and attitudes. Accordingly, this study confirmed the clinical competency of nursing knowledge, problem-solving abilities pursued through teaching and learning methods and design, and self-directed learning abilities. The third step is to determine whether learners are applying their trained knowledge, skills, and attitudes in the field and to evaluate their performance in the field. The fourth step is to evaluate the long-term effectiveness of learning, involving assessments related to organizational performance and contribution. However, this study provides fundamental data to suggest the direction of clinical practice guidance in nursing through the short-term effects of nursing practice education strategies. We propose a study that measures long-term effects to test the effectiveness of Kirkpatrick’s steps 3 and 4.

### 4.2. Effectiveness of the Nursing Practice Education Using M-Learning

Learning satisfaction, as confirmed by the first-stage response evaluation, was higher in the experimental group than in the control group; however, the difference was not significant. These results are similar to those of a practical training study [37] that applied smartphone videos to basic nursing skills. Meanwhile, it differed from a study in which nursing students who participated in a mobile-based nursing competency evaluation system [38] and smartphone application infant airway obstruction practice education [39] had high satisfaction with the practice.

Students’ learning satisfaction with clinical practice is influenced by the clinical environment [40]. The ward manager’s leadership, the pedagogical environment of the ward, and the role of field leaders are considered crucial [40]. According to students, the frequency of meetings with field instructors significantly enhances their satisfaction with the clinical practice environment [40,41]. It is unfortunate that neither the experimental nor control groups received any intervention related to changes in the clinical practice environment in this study.

The difference in results for learning satisfaction could be attributed to the absence of changes in direct nursing performance opportunities and the direct education of clinical field educators. Essentially, there was a change only in the role of the professor in clinical practice education and in the learner role of students, with no significant change in the role of the leader in the clinical field. Practical training can enhance learning satisfaction in clinical training by providing active guidance for clinical practitioners.

Examining studies related to learners’ learning satisfaction according to teaching and learning methods, it was found that innovative teaching methods do not affect learning satisfaction, independent of learning achievement [42]. What needs to be considered when applying a new teaching method is the interaction between the instructor and the learner [43]. In this study, effective interactions between students and instructors were considered; however, interactions with clinical instructors were not considered. Clinical field instructors are educators that nursing students most frequently encounter during clinical practice, making their role a significant factor. Future nursing practice education strategies need to include a clear presentation of the role of field instructors in clinical practice education and an agreement on the contents of education.

Clinical competency in the second stage of the learning evaluation was significantly higher in the experimental group than in the control group. Looking at the results of detailed subfactors, there was the biggest difference in the field of “nursing skill”. It is believed that the process of analyzing specific learning through examples or observations in the clinical field in the m-learning program and linking them with theoretical learning content served as reinforcement of learning. In addition, the lack of cognitive presence, which has been pointed out as a disadvantage of Internet-based but online-mediated learning, is believed to have been compensated for in this study’s strategy.

The researcher’s scenario was used to evaluate students’ skills and scores for clinical performance evaluation in this study. Self-reporting by learners has been the most common method for evaluating clinical performance in previous studies [44]. In the future, methods that objectively measure students’ technical skills should be developed. In addition, it is necessary to select skills and nursing activities that could strengthen field practice abilities during clinical practice so that learning can be conducted through the stages of theoretical learning, field observation, and actual performance.

Problem-solving ability, which was the second learning evaluation, was significantly higher in the experimental group. The problem-solving process involves problem discovery, problem definition, problem-solving design, problem-solving implementation, and problem-solving review [45]. It is believed that the experimental group experienced the problem-solving process, as several questions were raised based on their experience in the process of collecting information, analyzing applied cases, and linking them to observed nursing activities. Considering various nursing education intervention studies that improved problem-solving skills, one thing they had in common was that students took the lead in learning by actively using the examples and cases provided [46]. Improving problem-solving abilities is not the only factor. It requires experience in actively analyzing the process by applying it to examples or cases; therefore, it is necessary to actively reflect this in the nursing practice curriculum.

Finally, there was no significant difference in self-directed learning abilities between the two groups. Looking at previous studies that applied various curricula to nursing education, the effects on self-directed learning abilities were inconsistent [47]. The factors that influence these conflicting results are that self-directed learning requires a sense of ownership, autonomy, activity, and responsibility for learning [48]. In the experimental group, the time they had to learn on their own increased as they added m-learning training, compared to the actual control group. Consequently, m-learning education comes across as a burden to students and is considered mandatory, which may dilute their learning motivation and independence. This result is similar to that of a previous study [49], which found that most nursing students had positive attitudes toward m-learning, although there were some technical difficulties and burdens. Therefore, to expect changes in self-directed learning abilities, it is necessary to provide time for immersion in learning and situational considerations to stimulate and maintain learning motivation.

Another reason why different studies have produced conflicting results is that self-directed learning ability is a habit formed in the long term. A study on web-based surgical nursing learning contents [50] also showed nonsignificant results regarding self-directed learning ability and explained that the related factor was that it took a long time for self-directed learning ability to improve. Therefore, although the nursing practice education strategy may have an improvement effect on the direct knowledge of related subjects or clinical performance ability, even in a short period, it would be helpful to evaluate self-directed learning ability, which acts as an individual’s study habit, after a long period of education and training. Therefore, we recommend further research to improve self-directed learning abilities through long-term education.

Nursing students receive clinical practice instruction at medical sites from field instructors, but real-time guidance is not consistently provided by instructors at these locations. Therefore, continuous development of teaching and learning methods and media is necessary to achieve learning goals. Online learning is one such method, allowing for convenient learning without time and space constraints. It can be regarded as a technique that facilitates self-directed learning by permitting repeated learning based on the learner’s proficiency level [51]. Although this study was conducted prior to COVID-19, online learning remains active post-COVID-19, with increasing demand owing to limitations in the clinical environment [51,52]. To address the limitations of online learning, this study recommends blended learning, which combines online learning with in-person learning, rather than relying solely on online learning [53]. This study proposes m-learning clinical content using a tablet PC as a strategy to enhance the effectiveness of clinical practice education for nursing students. This is believed to provide foundational data for future utilization of teaching and learning methods and media in nursing education.

### 4.3. Limitation

A limitation of this study is that causal reasoning cannot be stronger than in true experimental design studies because this study used a nonequivalent control-group posttest design. Therefore, attention should be paid to the interpretation of research results. Students were recruited only from certain schools. Each school had a different practical curriculum, and there may be other exogenous variables; therefore, attention should be paid to the interpretation. The researcher selected a specific population of nursing students at D University to receive a three-week educational program as part of their curriculum. Preventing the spread of experimental treatments was beneficial, but the effects could only be seen for a short period of time. We propose measuring and evaluating this program after a semester in the future.

## 5. Conclusions

The effectiveness of a m-learning program created using tablet PCs based on the ADDIE model was evaluated in this study. A teaching and learning guidance plan was created and operated by the m-learning program in accordance with Gagné’s nine instructional events. To understand the effectiveness of the nursing practice education strategy, a two-stage evaluation was conducted using Kirkpatrick’s four-stage evaluation model. As a result, nursing students who participated in nursing practice education using m-learning improved their clinical performance and problem-solving abilities; however, there was no significant difference in learning satisfaction or self-directed learning ability.

These results appear to have influenced learning satisfaction due to the overload of learning content, the role of clinical field instructors that did not change, and the lack of opportunities to perform direct nursing. In addition, in the case of self-directed learning ability, it was analyzed that the heavy amount of learning, the time required for learning, adaptation to new class management, and the burden of learning in parallel with practice somewhat diluted the learner’s independence. In addition, self-directed learning ability is similar to study habits, so it may take time to change. To prepare a nursing practice education strategy, it is important to take into account the elements identified in this study. Furthermore, it is expected that continuous m-learning operations and development plans will be prepared and utilized in the operation and guidance of nursing practices.

Further testing and repeated research are suggested to test the long-term effects, as continuous follow-up evaluation of the program effect was not possible in this study. We suggest expanding research on various teaching and learning methods, media, and instructional designs related to nursing practice education. The development of teaching and learning guides will ensure consistency in practical education and lead to efficient classroom operations. Finally, an objective scale should be developed to evaluate nursing students’ clinical performance.

## Figures and Tables

**Figure 1 healthcare-12-00206-f001:**
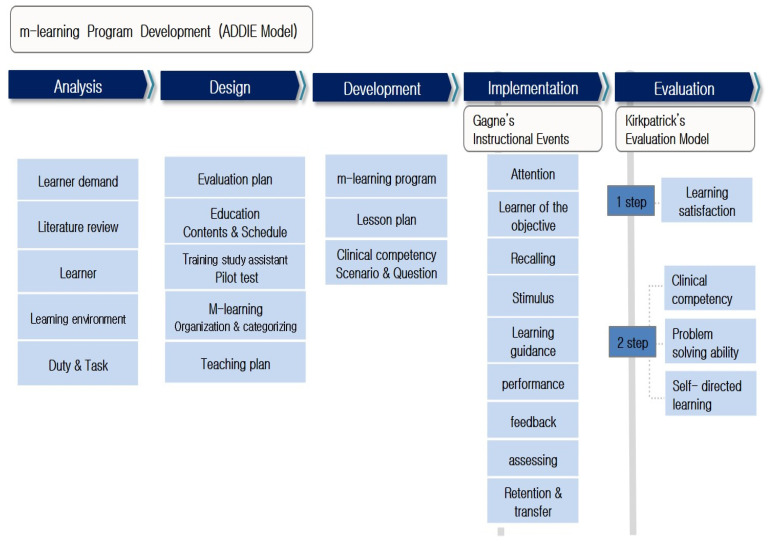
Development process of nursing clinical practice education with M-learning.

**Table 1 healthcare-12-00206-t001:** Research design of the study.

Group	Intervention	Posttest	Intervention	Post-Test
Cont.	X1	c1		
Exp.			X2	e1

Con. = Control group; Exp. = Experimental group; X1 = Traditional clinical nursing practice; X2: M-learning nursing practice education with a tablet PC; c1 = post-test of the control group; e1 = post-test of the experimental group.

**Table 2 healthcare-12-00206-t002:** Usage of table PC in m-learning.

Time	Usage of Tablet PC
Before clinical practice	- Check self-learning level - Learning nursing skill via video - Pre-learning test
In clinical practice	- Performing assignments - Learning ward key nursing intervention via video - Q&A - Providing clinical practice guide information
After clinical practice	- Quiz - Q&A - Guiding related extended learning contents

**Table 3 healthcare-12-00206-t003:** Main contents of clinical practice education using m-learning.

Week	Program Title	Learning Content
1st week	- Preparing patient care - Identifying the target treatment environment	- Ward orientation - Key terms and abbreviations - Takeover - Major health problems - Diagnostic tests & procedures - Treatment & nursing
2nd week	- Improving professional competency	- Basic nursing skills - Medical equipment - Emergency department - Patient safety - Infection control - Coordination of interdisciplinary - Legal & ethical standards - Recent trends in nursing care
3rd week	- Assessment of patient’s needs - Understanding the nursing documents - Nursing care for patient	- Identify patient health problems - Diagnostic tests & results - Treatment process - Applying nursing process according to major nursing problems - Make an education plan - Cooperation with other departments

**Table 4 healthcare-12-00206-t004:** Main contents of professor activities by Gagné’s nine instructional events.

Stage	Instructional Events	Contents/Activities	
1	Gaining attention	- Greetings - Asking questions - Watching video	
2	Informing the objectives	- Presenting instructions and m-learning - Reading learning topics - Recognizing related nursing works	
3	Stimulation recall of prior learning	- Checking m-learning review - Guiding related learning contents - Providing feedback on assignments - Checking learning status	
4	Presenting stimuli with distinctive features	- Presenting m-learning cases - Selection specific subjects and patients - Presenting learning topics	
5	Providing learning guidance	1st week	- Preparing nursing care for patients - Knowing patients’ environment
		2nd week	- Improving professionals’ competency
		3rd week	- Assessing patients’ demand - Analyzing meaning of documents - Nursing patient of case study
6	Eliciting performance	- Assignments - Answering questions - Presentation & discussion - Performing in a laboratory	
7	Providing informative feedback	- Practice questions - Peer evaluation & faculty feedback - Instructions on clinical field	
8	Assessing performance	- Group activities & test - Edit assignments, clinical observation - Reaffirming learning goals	
9	Enhancing retention and learning transfer	- Re-learning - Re-observation on clinical field - Applying to patient of case study	

**Table 5 healthcare-12-00206-t005:** Homogeneity test for general characteristics.

Characteristics	Categories	Exp. (n = 21)	Con. (n = 21)	χ^2^ or t	*p*
		n (%) or Mean (SD)	n (%) or Mean (SD)		
Gender	Female	19 (90.5)	21 (100.0)	2.10 *	0.488
	Male	2 (9.5)	0 (0.0)		
Age (years)	≦21	16 (76.2)	11 (52.4)	7.09	0.281
	≧22	5 (23.8)	10 (47.6)		
Grades	≧4.0	2 (9.5)	3 (14.3)	0.23 *	1.00
	3.0~3.9	18 (85.7)	17 (81.0)		
	<3.0	1 (4.8)	1 (4.8)		
Adult nursing score	≧90	6 (28.6)	4 (19.0)	1.07 *	0.801
	80~89	12 (57.1)	12 (57.1)		
	70~79	2 (9.5)	4 (19.0)		
	60~69	1 (4.8)	1 (4.8)		
Satisfaction with major of nursing		6.10 ± 1.55	7.33 ± 2.22	2.10	0.052
Interest with major of nursing		6.29 ± 1.49	6.91 ± 2.07	1.11	0.273

Con. = Control group; Exp. = Experimental group; SD = Standard deviation. * Fisher’s exact test.

**Table 6 healthcare-12-00206-t006:** Effects of nursing practice education using m-learning.

Characteristics	Categories	Exp. (n = 21)	Con. (n = 21)	t	*p*
		Mean (SD)	Mean (SD)		
Learning satisfaction		3.98 (0.52)	3.91 (0.59)	0.37	0.711
Clinical competency		2.87 (0.33)	2.12 (0.32)	7.44	<0.001
	Nursing process	2.79 (0.45)	2.06 (0.40)	5.62	<0.001
	Nursing skill	3.26 (0.41)	2.11 (0.38)	9.49	<0.001
	Cooperation	2.46 (0.28)	1.76 (0.49)	5.73	<0.001
	Communication	3.13 (0.49)	2.62 (0.86)	2.36	0.023
	Professionalism	2.68 (0.54)	2.37 (0.27)	2.39	0.022
Problem-solving ability		3.74 (0.41)	3.44 (0.43)	2.28	0.028
Self-directed learning		3.39 (0.44)	3.18 (0.32)	1.68	0.101

## Data Availability

The data presented in this study are available on request from the corresponding author. The data are not publicly available due to privacy concerns.

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
