# Peer review of "Development and Evaluation of Nursing Clinical Practice Education Using M-Learning"

_healthcare, 2024, doi:10.3390/healthcare12020206_

Round 1

Reviewer 1 Report

Comments and Suggestions for Authors

1. It is an interesting and relevant subject for nursing education

2. In methods, measuremets (2.4.) scales could be a litle more described (2.4.1., 2.4.2., and so on)

3. Analysis is vague in the begining - "five nursing students at D University who had completed clinical practice, 158 in line with a previous study [29]." ou "literature review was conducted using five databases in Korea." (?)

4. Could be improved the relaion wioth, for example, "applying the Kirkpatrick evaluation model"

5. Discussion is extense. However, some outcomes are not discussed (for example, the Learning satisfaction in both groups (only related with teaching and learning methods)

6. Conclusion is filled with recommendations, coulde be improved as a synthesis. 

Author Response

Thank you for each reviewer’s thoughtful review and comments. Each comment is very useful and constructive for this paper. I deeply appreciate all reviewers. According to Healthcare reviewers’ comments, I revised some parts of the manuscript and presented them using table below. The changed one in manuscript marked with Red color.

Reviewer 2 Report

Comments and Suggestions for Authors

To authors, 

This is a good evidence-based study that demonstrates the effectiveness of m-learning to improve clinical competence.

1)   A shortcoming of this study is that it was a non-equivalent control group post hoc design and did not compare the pre-homogeneity of the two groups, so caution should be exercised in its interpretation. Nevertheless, the finding of improvement in clinical competency and problem-solving ability in the group that performed m-learning after the intervention may provide evidence for the possibility of expanding the application of various teaching methods in the future.

2)   Self-directed learning was not significantly different between the experimental and control groups, but there is an error in the description of the results in the text. Please correct.

3)   In section 2.5.4 Implementation, please consider adding a description of instructor activities consistent with Gagne's guidelines as text in addition to the table. This could be tied to the instructor roles presented in the Discussion section.

I wish you all the best in your revision work and hope to see this manuscript as a published research article.

Sincerely, your reviewer

Author Response

(The authors gave the same response as above.)

Reviewer 3 Report

Comments and Suggestions for Authors

Dear All,

It was with great interest that I read the paper “Development and evaluation of nursing clinical practice education using m-learning”. The paper is an extremely valuable contribution to
the scholarly discussion about nursing practice education. Having analysed the article, I reached
the following conclusions:

1.      The content of the work is consistent with the aim of the work.

2.      The Introduction section is exhaustive, however, it might be wise for the authors  to rewrite the aims of the study in such a way so that there is one main aim of the study.

3.      The Methodology section should be further developed.

4.      The Results section was written in a comprehensible way.

5.      The Discussion section does not contain direct references to all indicated aims of the work.

6.      Conclusions are correct.

7.      The research literature was correctly selected.

What may raise some doubts is the small size of the student group participating in the study and
the lack of detailed information on the exclusion and inclusion criteria, as well as the lack of data on the entire study population (apart from the references to previous reports and how the study population was statistically calculated). The research material was collected in 2016 (almost 8 years ago). As a result of the COVID pandemic and the conditions of providing academic education at that time, it became outdated.  If the authors do not believe that the collected data is outdated, they should elaborate on that. Another fundamental problem with the study is connected with the time period for evaluating the effects of the said study – it is not clear whether three weeks of using
m-learning would allow to fully evaluate its effectiveness.

Author Response

(The authors gave the same response as above.)
